# Essential Oil—Loaded Nanofibers for Pharmaceutical and Biomedical Applications: A Systematic Mini-Review

**DOI:** 10.3390/pharmaceutics14091799

**Published:** 2022-08-26

**Authors:** Ioannis Partheniadis, Georgios Stathakis, Dimitra Tsalavouti, Jyrki Heinämäki, Ioannis Nikolakakis

**Affiliations:** 1Department of Pharmaceutical Technology, School of Pharmacy, Faculty of Health Sciences, Aristotle University of Thessaloniki, 54124 Thessaloniki, Greece; 2Institute of Pharmacy, Faculty of Medicine, University of Tartu, 50411 Tartu, Estonia

**Keywords:** electrospinning, emulsion, coaxial, wound healing, topical delivery, antimicrobial, long-term stability

## Abstract

Essential oils (EOs) have been widely exploited for their biological properties (mainly as antimicrobials) in the food industry. Encapsulation of EOs has opened the way to the utilization of EOs in the pharmaceutical and biomedical fields. Electrospinning (ES) has proved a convenient and versatile method for the encapsulation of EOs into multifunctional nanofibers. Within the last five years (2017–2022), many research articles have been published reporting the use of ES for the fabrication of essential oil—loaded nanofibers (EONFs). The objective of the present mini-review article is to elucidate the potential of EONFs in the pharmaceutical and biomedical fields and to highlight their advantages over traditional polymeric films. An overview of the conventional ES and coaxial ES technologies for the preparation of EONFs is also included. Even though EONFs are promising systems for the delivery of EOs, gaps in the literature can be recognized (e.g., stability studies) emphasizing that more research work is needed in this field to fully unravel the potential of EONFs.

## 1. Introduction

Essential oils (EOs) are complex mixtures of volatile compounds synthesized by plants for defense and signaling purposes [1,2]. EOs exhibit also a broad spectrum of biological properties such as antimicrobial, analgesic, anti-inflammatory, antidiabetic, antitumor at the same time, etc. [3,4,5]. The increasing problem of bacterial resistance to antibiotics has led researchers to the quest for new antimicrobial agents. EOs are characterized in general by accepted safety and efficacy profiles, and therefore they may be promising candidates against microbial infections [1,6,7]. EOs are widely studied for their pharmaceutical applications and recent studies have dealt with their formulation into pharmaceutical solid dosage forms such as tablets [8] with targeted delivery to specific sites of the human body [9]. In addition, EOs hold promise as antimicrobial agents for topical wound healing applications [10]. However, the use of EOs in the pharmaceutical and biomedical fields is still limited due to several drawbacks. EOs are volatile, have strong aromas and a bitter taste, are easily oxidized and in general, they are chemically unstable. These drawbacks are still understudied and hence formulation challenges remain [11,12,13].

Encapsulation via spray drying, coacervation, ionic gelation or electrospinning (ES) in micron-, submicron- or nano- scale systems is a widely applied strategy in food, polymer and pharmaceutical industries pertinent to EO formulation [14,15,16,17]. Encapsulation of EOs can improve their physicochemical stability and organoleptic characteristics and provide controlled/targeted release profiles [7,8,9,18]. Moreover, encapsulation in nanoscale systems could improve bioavailability as a result of improved dissolution in biological liquids and permeation through biological barriers, thus enhancing their biological properties [19,20]. Over the various encapsulation techniques available, ES provides important advantages: (i) it can produce very thin fibers (of few nanometers in diameter) with large specific surface areas (high surface-to-volume ratio) and large interstitial spaces (voids); (ii) it can provide ease of functionalization; (iii) the produced electrospun nanofibers exhibit superior mechanical properties; (iv) it is a versatile process and (v) it is amenable for scale-up [21].

The study of electrospun nanofibers loaded with essential oils (EONFs) arose in the early 2010s aiming mainly at nutraceutical products (e.g., active food packaging) [22]. However, in the last five years (2017–2022) the fabrication of EONFs has gained great interest and attracted many researchers to work in this field. Encapsulation of EOs into polymeric nano/microfibers has enabled new approaches to wound dressings, scaffolds for tissue engineering and controlled/targeted delivery thereof [23]. Controlled release of EOs from nanofibers can also minimize the cytotoxic effects of some EO components on human cells [22,23]. Moreover, EOs can attain stability in the EONF systems and thus render prolongation of product’s shelf-life, thus opening the way to commercialization.

The present review article aims to give an overview of the pharmaceutical and biomedical applications of essential oil—loaded nanofibers (EONFs) and demonstrate their advantages over traditional polymeric films loaded with EOs. ES technologies utilized for the encapsulation/solidifying of EOs are briefly reviewed. The polymers that are used for the preparation of EONFs are also discussed. Finally, the challenges and outlooks of these systems are presented in an attempt to provide future perspectives on the topic.

## 2. Methods

The present review was conducted according to the PRISMA systematic review statement guidelines [24,25]. The flow diagram describing the selection, identification and screening methodology of the studies of interest included in this review article, is presented in Figure 1.

### 2.1. Electronic Resources/Bibliographic and Full Text Databases

For the search strategy, the following databases were used: (i) Scopus^®^ (Elsevier) and (ii) PubMed^®^ (NLM: United States National Library of Medicine). The searching criteria included the following two main keywords: “essential oil(s)”, and “nanofibers”. In a second step, the results from the aforementioned searching criteria were limited including the following keywords (each one at a time): “pharmaceutics”, “delivery”, “biomedical” and “wound healing”.

### 2.2. Study Selection

The first step of the studies selection strategy described above led to the identification of 314 documents by Scopus^®^ and 106 documents by PubMed^®^. Furthermore, the additional keywords—used in the second step of the search strategy—reduced the number of resulting documents to 37 and to 15 for Scopus^®^ and PubMed^®^ databases, respectively. After excluding duplicates and irrelevant documents (e.g., articles that were not in English, out-of-topic articles dealing with food packaging films, etc.), a total of 26 studies remained. Furthermore, a secondary search was performed in the reference lists of the identified studies to detect research papers, which did not appear in the database search but fulfilled the acceptance criteria. Hence, five more studies were identified and included in the present review resulting in a total number of 31 studies.

## 3. Electrospinning of Essential Oils

Electrospinning (ES) technology has been extensively explored and documented in the literature for the fabrication of multifunctional drug delivery systems [26,27]. Consequently, its potential use in the preparation of EONFs is justifiably explored. In the literature, two main ES technologies have been reported for the fabrication of EONFs: (i) the traditional ES set-up and (ii) its coaxial modification. For fabricating EONFs, both natural and synthetic polymers have been used as carrier materials. A wide range of actives (such as small molecular weight active pharmaceutical ingredients—APIs, biological materials, cells or bacteria, etc.) can be electrospun using liquids (such as melts), solutions or suspensions as solvent systems [21,28]. For the fabrication of EONFs, either spinning solutions or emulsions have been utilized. In this section, a brief overview of the set-up and working principle of ES, and the main carrier polymers used for preparing EONFs intended for pharmaceutical and biomedical applications, will be presented and discussed.

### 3.1. Overview of Electrospinning Set-Ups Utilized for the Fabrication of EONFs

#### 3.1.1. Conventional Electrospinning (ES)

A conventional ES experimental set-up consists typically of four main parts: a high-voltage power supply; a spinneret (usually a metallic needle) fitted to a syringe; a syringe pump, and a grounded collector (usually a copper or aluminum plate) [27,29]. A schematic representation is presented in Figure 2. The high-voltage power supply (typically in the range of 5 to 25 kV [21]) is attached to a spinneret and collector, usually charging positively or negatively the spinneret and oppositely charging the collector. The collector is preferably grounded to enhance the deposition of the newly fabricated nanofibers on its surface [27]. The syringe receives feed liquid (usually a solution) at a constant flow rate controlled by a syringe pump, which subsequently forms a pendant droplet at the tip of the spinneret.

The high voltage applied to the spinneret causes the accumulation of electric charges on the droplet surface, where at some point a critical voltage is reached. When this critical voltage value is reached, the generated electrostatic repulsion overcomes surface tension and viscoelastic forces, and the formation of a cone, known as the Taylor cone, is observed. Therefore, the high-voltage electrical field is essential for the ES process, otherwise, the feed liquid will be extruded from the syringe in the form of spherical droplets [30].

At the apex of the Taylor cone (Figure 2h), the feed solution emits an electrified jet which elongates (a process known as jet thinning; Figure 2i) and is accelerated towards the grounded collector plate, while at the same time the solvent evaporates (Figure 2j) inducing jet solidification. At the end of the process, solid nano/microfibers are deposited on the collector surface forming usually a non-woven fibrous mat [27,29,30].

The parameters affecting the ES process have been widely explored in the literature and discussed in depth in a number of review articles (e.g., [30,31,32,33]). These include the properties of spinning liquid (viscosity, molecular weight of the polymer, conductivity and volatility of the solvent system), the applied voltage, the pumping rate, the spinneret to collector distance, the environmental temperature and humidity, and the collector design and geometry [21]. Lack of control or non-optimization of the aforementioned variables can lead to the formation of beaded—or with other defects—nanofibers [30,34].

#### 3.1.2. Coaxial Electrospinning (CES)

Coaxial electrospinning (CES) is a modification of conventional ES entailing an arrangement of two or more feeding systems to synchronously spin multiple polymeric liquids from coaxial syringe capillaries used as spinneret [35,36]. CES can produce continuous coated or hollow nanofibers suitable for the encapsulation of EOs [37,38]. Since CES is a modification of the conventional ES technology, they are conceptually similar. They share a similar main set-up, where the quality and morphology of the electrospun nanofibers are governed by the same parameters [36,39,40]. However, in the case of CES, the spinneret shows a more sophisticated design and geometry to enable the simultaneous spin of multiple feed liquids.

The first CES set-up was introduced and developed by Sun et al. (2003) [37]. In this set-up, a core-shell needle was utilized as the spinneret, which was attached to a double-compartment syringe. The experimental set-up of the first developed CES device is presented in Figure 3. As seen in Figure 3, the main geometry of a CES spinneret is designed by the insertion of a needle or capillary (inner needle) into a concentric outer needle, which communicate with the core and shell feed reservoirs, respectively [36,40].

As described in the case of conventional ES, the application of a high-voltage electric field will result in a Taylor cone. However, in the case of CES, the formed cone consists of the core liquid surrounded by the wall/shell liquid. A schematic illustration of the formation of this compound Taylor cone is shown in Figure 4a. The real-time photos of jet initiation and thinning from a compound Taylor cone by voltage increase, are presented in Figure 4b. On its way to the collector, the jet experiences bending instability and follows a back-and-forth whipping trajectory, during which the two solvents evaporate, and complex nanofibers are formed [41]. As the compound Taylor cone remains stable, the core is uniformly embedded into the wall/shell, thus resulting in the formation of core-sheath nanofibers.

#### 3.1.3. Advantages of ES and CES EONFs over Traditional Polymeric Films

ES and CES technologies for the fabrication of EONFs have shown a number of advantages compared to the traditional polymeric films loaded with EOs [8,21,38,41,43,44,45,46]:Improved isolation of the physically and chemically unstable EO and minimization of the chances of decomposition under highly reactive environmental conditions;Better sustained/prolonged or targeted release characteristics;Reinforcing the elastic and oily nature of EOs to improve their mechanical properties (and thus enhancing e.g., their densification and compression into tablets);Ability to serve as scaffolds for biomedical applications in which the less biocompatible EO is efficiently encompassed by a biocompatible polymeric material.

### 3.2. Spinning Emulsions in the Fabrication of EONFs

The description of the conventional ES and CES until this point was made assuming that the feed liquid was a polymeric solution. However, for the preparation of EONFs, both solution and emulsion ES have been used. As already discussed, in solution ES, the solidification of electrospun material(s) is based on rapid solvent evaporation. This is associated with some limitations related to solvents’ toxicity, environmental concerns and additional solvent extraction processes [47]. Moreover, spinning solutions in ES have been reported to show low production yield compared to melt ES [47,48]. Many strategies have been proposed to deal with the low productivity of solution—based ES, including multi-needle systems and needleless ES systems [21,47]. The approach of emulsion—based electrospun nanofibers was first reported by Sanders et al. (2003) [49] and this approach has recently drawn the attention of some research groups for the preparation of EONFs.

Emulsion-based ES (EES) technology involves the same basic set-up as conventional/solution or coaxial ES. The major difference, however, is that in EES a synchronous spinning of two immiscible solutions is implied. The fiber-forming polymer is dissolved in aqueous solvent(s) to form a continuous phase (O/W emulsions), while the EO is either solubilized in organic non-polar solvent(s) or used as such to form the internal phase. During EES, the continuous phase evaporates rapidly (as in solution ES) resulting in a viscosity increase. Thus, the internal phase droplets will migrate to the center of the jet. The droplets are then merged due to the mutual dielectrophoresis resulting in core-sheath structured nanofibers [50]. EES is more complex compared to ES due to the requirements of special spinnerets. Solvent and surfactant selection in EES is more time-consuming than in ES, since the preparation of physically stable and spinnable emulsions is critical. The type (ionic or non-ionic) and concentration of the solvent(s) and surfactant(s) are expected to affect the surface tension and the conductivity of emulsion, and ultimately the topology and the internal architecture of the EONFs [51].

It is evident that the generation of nano/microfibers by EES leads to the same nanofiber structures as in coaxial ES (CES). However, compared to CES, EES may damage the EO due to the interfacial tension between the two immiscible phases of the emulsion [47,52]. In Figure 5, a schematic illustration of spinnerets loaded with different spinning liquids for (i) conventional (traditional) ES, (ii) CES and (iii) EES is presented to better visualize the differences between the three methods utilized for the preparation of EONFs.

### 3.3. Biocompatible Polymers Used for the Preparation of EONFs

The studies included and discussed in the following section have utilized a wide range of polymers for the encapsulation of the EOs. Some of the polymers used in these research works are of natural origin, but most of them are synthetic or semi-synthetic. These polymers are also biocompatible, and they have shown high encapsulation efficiency for bioactive compounds. Moreover, their solutions are readily spinnable and for some natural polymers, methods to enhance their solution spinnability are presented. The polymers are presented in alphabetical order in the following sub-sections. Further to the presented polymers here, any other biocompatible, biodegradable and easily spinnable polymers can be used for the preparation of EONFs (e.g., polylactic acid, starch, whey protein isolate, methyl cellulose).

#### 3.3.1. Cellulose Acetate (CA)

Cellulose acetate (CA) is an esterified derivative of cellulose, which offers superior properties for ES compared to other available cellulosic derivatives [53]. Cellulose itself is insoluble in most of the common solvents due to the presence of inter- and intramolecular H-bonds (resulting in gel structures), which makes it challenging to be used in ES. Unfortunately, the solvents in which cellulose is soluble give solutions that are not spinnable [54]. Acetone is an exception, enabling the fabrication of cellulosic nanofibers. Moreover, ionic liquid solutions of CA were able to improve spinnability.

#### 3.3.2. Chitosan (CHS)

Chitosan (CHS) is the deacetylated derivative of chitin, which is the second most abundant polysaccharide after the similarly structured cellulose. CHS is soluble in organic acid solutions as well as in water, ethanol and acetone in the presence of a small amount of acid. CHS solutions are highly viscous, due to their amino groups that are positively charged in the pH range of 2 to 6 [55], which is a limitation for its use in ES [56]. Moreover, the formation of H-bonds in a 3D network restricts its chains’ movements in the electrical field applied during the ES process [57,58]. Thus, the selection of the right grade of CHS with suitable molecular weight and solution viscosity, along with suitable solution concentration is critical and needs to be optimized [56]. Mixing CHS with a variety of synthetic polymers, metal nanoparticles, metal oxides, zeolite or organic metal structures, could also enhance its solution properties and spinnability [59].

#### 3.3.3. β-Cyclodextrin (βCD) Derivatives

Beta-cyclodextrin (βCD) is a cyclic oligomer of glucopyranose structured as a hollow truncated cone. Its interior shows a partially hydrophobic character while its exterior is hydrophilic due to the presence of hydroxyl groups [60,61]. Among the different cyclodextrin grades, βCD is the less water-soluble one [62]. To overcome this limitation and enhance its spinnability, βCD has been chemically modified (e.g., methylated-βCD and 2-hydroxypropyl-βCD) [60,61,63]. An interesting aspect of βCD is that it can generate polymer-free electrospun nanofibers, thus avoiding the necessity to use high molecular weight polymers that will traditionally enable entanglements to ensure the formation of defect-free nanofibers [61].

#### 3.3.4. Gelatin (GEL)

Gelatin (GEL) is the hydrolyzed derivative of collagen and it is a polyampholyte protein [64,65]. Since GEL is derived from collagen, it shows good mechanical properties due to its peptide composition and mimicry [66]. GEL contains arginine-glycine-aspartate (RGD) sequences providing suitable sites for cell attachment [67]. However, GEL shows poor solubility in water forming a gel structure through strong intra- and intermolecular interactions between the polypeptide chains. To overcome this limitation and enable the spinning of its solutions, GEL is usually employed after hydration with hot water using specific ES set-ups, that are able to circulate hot water during the solution feeding process [68,69]. Moreover, to prevent gelation and facilitate electrospinning, GEL can be dissolved in fluorinated alcohols or acidic organic solvents prior to application.

#### 3.3.5. Gellan (GLL)

Gellan (GLL) is a natural exopolysaccharide produced from the bacteria *Sphingomonas elodea* [70]. GLL shows a complex sol–gel behavior while its aqueous solubilization is characterized as non-typical suggesting difficulties in ES processing [71]. Unstable Taylor cones of GLL solutions in ES have been also reported, and these drawbacks were assigned to the anionic nature, low shear viscosity and strong shear thinning behavior (at low shear rates) of such solutions [72,73].

#### 3.3.6. Polyacrylonitrile (PAN)

Polyacrylonitrile (PAN) also known as polyvinyl cyanide is a synthetic organic polymer resin and the precursor for high-performance carbon fiber, due to its ladder-like structure [74]. PAN has drawn attention due to the high carbon yield and mechanical properties of the produced carbon fibers [75]. It is widely explored in the electrochemical field due to its low electrical conductivity and insulation ability [74,76]. Researchers in this field have utilized PAN as a filament-forming polymer [77,78]. However, its use in the pharmaceutical and biomedical fields remains as an adsorbent of microorganisms, while its use in ES is not common due to its strong static electricity [79].

#### 3.3.7. Poly-ε-caprolactone (PCL)

Polycaprolactone (PCL) is a semi-crystalline linear aliphatic polyester offering combination of polyolefin-like mechanical properties and polyester-like hydrolysability [80]. Its rheological and viscoelastic properties allow its use in the ES process [81]. The hydrophobicity of PCL nanofibers demands modifications to enable higher biocompatibility and hydrophilicity, such as surface coating, plasma treatment, poly(dopamine) treatment, blending with the copolymer, alkali treatment and polymer grafting, making it favorable for a range of biomedical applications [82,83].

#### 3.3.8. Polyethylene Oxide (PEO)

Polyethylene oxide (PEO) is a polyether available in a wide range of molecular weights. MWs < 100,000 are generally called polyethylene glycols (PEGs), whereas higher molecular weight polymers are classified as PEOs [84]. PEOs are amphiphilic and readily dissolve in water and in a variety of organic solvents [85,86]. Due to their high MW, PEOs have been utilized extensively for the encapsulation of small molecules.

#### 3.3.9. Poly(L-lactide-co-ε-caprolactone) (PLCL)

Poly(L-lactide-co-ε-caprolactone) (PLCL) is a copolymer of polylactic acid (PLA) and polycaprolactone (PCL) in a mass ratio of 50:50 and it is usually characterized as a hydrophobic aliphatic polyester copolymer [87,88]. It is often applied as a mechano-stimulating tissue engineering scaffold [89,90,91]. However, its surface lacks adhesive and structural proteins that would enable cell adhesion, proliferation and tissue remodeling [92]. Hence, it has been combined with polymers of natural origins (e.g., collagen) to enhance its biological properties [93].

#### 3.3.10. Polyurethane (PU)

Polyurethanes (PUs) contain the urethane group (–NH–(C=O)–O–) in their structure and are thermosetting polymers [94]. They are capable of strong intermolecular bonding and hence they have been utilized for applications in adhesives and coatings, elastomers, foams and tissue engineering [95]. Electrospun PU nanofibers exhibit good mechanical and adhesion properties and have been used as wound dressing materials in biomedical applications and drug delivery [96].

#### 3.3.11. Polyvinyl Alcohol (PVA)

Polyvinyl alcohol (PVA) is a non-ionic hydrolysis derivative of polyvinyl acetate (PVAc) and it is a highly hydrophilic semicrystalline polymer with excellent mechanical properties [97]. However, PVA itself lacks bioactivity, and in order to produce functional nanofibers, blending with natural polymers or biomolecules is usually required to produce electrospun nanofibers with accelerated wound healing characteristics [98]. PVA shows thermal stability, high viscoelastic properties, good chemical stability and the ability to enhance the mechanical properties of the electrospun nanofibers [99,100].

#### 3.3.12. Polyvinylidene Fluoride (PVDF)

Polyvinylidene fluoride (PVDF) is the polymerization derivative of vinylidene difluoride. PVDF is a highly non-reactive thermoplastic and electroactive polymer [101]. Due to its polar crystalline nature, it is able to produce large voltages with low forces and therefore it is mainly utilized for piezoelectric applications [102,103]. Nevertheless, its applications in the biomedical field are also recognized since it has shown good cell adhesion properties and long-term stability, e.g., for the preparation of multifilament for vascular grafts, ligament and artificial cornea [104] and for smart piezoelectric biomaterials [105,106].

#### 3.3.13. Polyvinyl Pyrrolidone (PVP)

Polyvinyl pyrrolidone (PVP) is the polymerization derivative of N-vinylpyrrolidone. PVP has unique properties of aqueous solubility, as well as in many organic solvents [107,108]. Its water-affinity and good adhesion properties make it one of the most useful materials in the biomedical and pharmaceutical fields [109]. PVP has shown high potential as wound-dressing material since it maintains wound-moisturization preventing dehydration and scab formation [110]. It has also been extensively used as an excipient in a variety of drug-delivery systems [111].

#### 3.3.14. Silk Fibroin (SF)

Silk fibroin (SF) is one of the two types of proteins (the other one is sericin) in the silk, that is produced by various insects including the silkworm. SF forms the filaments of silkworm silk and can be regenerated in various forms, such as gels, powders, fibers or membranes [112]. SF shows several distinctive biological properties, such as good oxygen and water vapor, permeability and minimal inflammatory reaction [113,114]. However, SF shows low hydrophilicity and low mass loss rate, limiting its use in ES [67].

#### 3.3.15. Sodium Alginate (SA)

Sodium alginate (SA) is a natural polymer extracted mainly from brown algae. Chemically it is a linear polysaccharide derivative of alginic acid comprised of 1,4-β-d-mannuronic and α-l-guluronic acids. It shows high hydrophilicity with good aqueous solubility. It has antibacterial properties, and it is non-immunogenic, which explains its wide use in the pharmaceutical field [115,116]. However, ES of SA is a challenging process, due to its poor solubility in organic solvents, high conductivity and surface tension [117].

#### 3.3.16. Zein (ZN)

Zein (ZN) is a group of prolamine proteins found in the endosperm of maize (*Zea mays*). It shows hydrophobic characteristics (due to the presence of leucine and alanine in its structure), low water vapor permeability and greaseproof properties [118,119]. The ES processing of ZN is challenging due to the frequent clogging of the spinneret when aqueous/ethanol solutions are used and the necessity of using hazardous solvents as alternatives. There are also limitations in reusing its solutions [120].

## 4. Pharmaceutical and Biomedical Applications of Essential Oil-Loaded Nanofibers

The state-of-the-art studies on EONFs aiming at pharmaceutical or biomedical applications are summarized in Table 1. As seen in Table 1, most studies employ ES technologies based on conventional/solution set-ups (namely 74% of the total studies presented), rather than emulsion ES (16% of the total studies) and fewer studies used coaxial ES (10% of the total studies). Moreover, most of the prepared EONFs aim at biomedical applications and more specifically as wound dressings (52% of the total studies), which can be attributed to the good mechanical properties of the nanofibers and their presentation as non-woven mats.

Most of the studies listed in Table 1 are case studies, and only a small number attempted to systematically study the preparation of EONFs based on experimental design and optimization [121,122,123,124]. Furthermore, not all studies in Table 1 concerned EO formulations. Few of them utilized pure phytochemicals [125,126,127,128] and one of them used herbal extracts [129]. These studies, although out of the scope of the present review, are included in Table 1 since they provide some useful insights into the ES technologies and polymers used. To avoid misinterpretations, these studies are indicated in Table 1 by distinct marks.

In the following sub-sections, the most indicative studies (i.e., studies that were based on an experimental design method and fully characterized the prepared EONFs in terms of physicochemical changes, possible molecular interactions, mechanical properties, bioactivity, and thermal, chemical and physical stability) utilizing either conventional solution ES, EES or CES for the preparation of EONFs are discussed.

### 4.1. Studies Utilizing Conventional Solution Electrospinning (ES)

Conventional solution ES for the fabrication of drug-loaded nanofibers is a well-studied topic in the literature. Thus, it is the most explored technology for the preparation of EONFs among the ES technologies presented.

Tonglairoum et al. [130] utilized conventional solution ES to encapsulate betel or clove EO into PVP/hydroxypropyl-β-cyclodextrin (HP-βCD) nanofibers with antifungal properties for the prevention and treatment of denture stomatitis associated with *Candida albicans*. They studied the topology and size of the EONFs by SEM, possible chemical interactions by FTIR spectroscopy, thermal behavior by DSC and evaluated the EONFs for their mechanical properties. Moreover, they studied the prepared EONFs for in vitro release, antifungal activity and cytotoxicity potential. SEM images showed that EONFs had smaller diameters (between 397 and 426 nm, Table 1) compared to placebo nanofibers. FTIR spectroscopy and DSC studies confirmed encapsulation of the essential oils into the nanofibers. EONFs showed the fast release of EOs and immediate reduction of *Candida albicans* cells in vitro. In addition, the EONFs appeared to be safe when applied for a short time to the oral mucosa. Hence, the authors concluded that the EONFs can potentially be used as materials for oral hygiene maintenance in denture stomatitis for oral candidiasis prevention. The present results, however, should first be confirmed by in vivo studies.
pharmaceutics-14-01799-t001_Table 1Table 1Studies on essential oil—loaded nanofibers (EONFs) intended for pharmaceutical or biomedical applications.StudyEssential OilPolymer ^a^Solvent/Surfactant for EO ^b^Electrospinning Set-UpNanofiber Size (±SD)Characterization ^c^Stability StudiesApplicationRef.01LavenderPANDMF/n.a.Conventional/solution89–143 nm(n.d.)TGA, FTIR, LE, in vitro antibacterial and cytotoxicity 30-daysantibacterial activityAntimicrobial agent delivery[79]02Cinnamon, lemongrass, peppermintCAAcetone/n.a.Conventional/solution0.9–2.8 μm(0.3–1.1)Raman, in vitro antibacterial and proliferation n.d.Antibacterial wound dressings[131]03LavenderSA, PEODMF/Pluronic^®^ F127Conventional/emulsion93 nm(22)Raman, wettability, mechanical properties, in vitro release, antibacterial, cytotoxicity and anti-inflammatory, in vivo cytokine expressionn.d.Antibacterial/anti-inflammatory burn dressings[132]04Betel, clovePVP, HP-βCDWater, ethanol/n.a.Conventional/solution397–426 nm(65–82)FTIR, DSC, ex vivo mucoadhesion, LE, in vitro release, in vitro antifungal and cytotoxicityn.d.Denture stomatitis prevent and treatment[130]05ThymeSF, GELFormic acid/n.a.Conventional/solution/soaking182–380 nm(23–49)FTIR, mechanical properties, porosity, contact angle, LE, WVTR, in vitro antibacterial, MTTn.d.Antibacterial agent delivery[67]06Clove, cinnamon, LevanderPVAn.a./SAConventional/emulsionn.d.FTIR, liquid absorption properties, in vitro antibacterialn.d.Antibacterial wound dressings[123]07Eugenol (phytochemical)PANDMSO/n.a.Conventional/solution179–218 nm(25–32)FTIR, in vitro release, in vitro antifungaln.d.Antimicrobial agent delivery[128]08ThymePVP, GELn.a./Cremophor^®^ RH 40Conventional/emulsion202–316 nm(48–127)FTIR, LE, in vitro antibacterial8-days antibacterial activity (24 and 37 °C)Antibacterial agent delivery[133]09*Ferula gummosa Boiss*ZNEthanol/n.a.Conventional/solution677–727 nm(26–58)FTIR, DSC, LE, in vitro release, in vitro antioxidant and α-amylase and α-glucosidase inhibition activityn.d.Anti-diabetic agent delivery[134]10PeppermintPUDMF/n.a.Conventional/solution359–997 nm(134–166)TGA, AFM, mechanical properties, in vitro APTT and PT assay, in vitro hemolysis, in vitro biocompatibilityn.d.Antibacterial and coagulant wound dressings[135]11OreganoPLCL, SFHFIP/n.a.Conventional/solution496–521 nm(136–150)FTIR, TGA, porosity, mechanical properties, in vitro release, in vitro antioxidant and anti-tumor activitiesn.d.Antioxidant and anti-tumor agent delivery [136]12LavenderPUDMF, THF/n.a.Conventional/solution639–979 nm(267–371)FTIR, XRD, wettability, in vitro antibacterial and cytotoxicity, in vitro cell attachmentn.d.Antibacterial wound dressings[137]13PeppermintPEOEthanol/n.a.Conventional/solution318–364 nm(104)FTIR, DSC, in vitro antibacterial, MTT, in vivo wound healingn.d.Antibacterial wound dressings[138]14ClovePCL, GELGAA/n.a.Conventional/solution285–305 nm(67–82)FTIR, LE, in vitro antibacterial, wettability, in vitro cell viability and wound healingn.d.Antibacterial wound dressings[139]15*Syzygium aromaticum*PANDMF/n.a.Conventional/solution141–143 nm (n.d.)FTIR, TGA, wettability, in vitro release, in vitro antibacterial and cytotoxicityn.d.Antibacterial agent delivery[140]16ClovePVAGMA, GAA/Tween^®^ 80Conventional/emulsion306 nm(92)FTIR, DSC, LE, ex vivo skin permeation, in vivo anti-inflammatory, skin irritation test6-month EO retention study (refrigeration and ambient) Anti-inflammatory agent topical delivery[121]17Carvacrol, cinnamaldhehyde, thymol, β-caryophyllene, squalene, tyrosol, curcumin (phytochemicals)PCLDCM, DMF/n.a.Conventional/solution258–948 nm(47–277)Water uptake, LE, in vitro release, in vitro cell viability and anti-inflammatory, immunofluorescence assaysn.d.Anti-inflammatory woundhealings[125]18*Myrocarpus fastigiatus*PVA, CHSn.a./SDSConventional/emulsion275–370 nm (n.d.)FTIR, DSC, in vitro release, in vitro antimicrobialn.d.Antimicrobial agent delivery[141]19Tea tree, cinnamon, clovePUDMF, THF/n.a.Conventional/solutionn.d.Mechanical properties, in vitro antimicrobial and cytotoxicityn.d.Antimicrobial agent delivery[142]20ThymeZN, PEOWater, ethanol/n.a.Conventional/solution6.1 μm(0.6)FTIR, contact angle, in vitro antibacterial, in vivo wound healingn.d.In situ antibacterial wounddressings[143]21Oregano, turmeric (extracts)PVAWater/n.a.Conventional/solutionn.d.In vitro antibacterial and antioxidant, MTT, in vivo histologicaln.d.Diabetic ulcer wound treatment[129]22*Satureja mutica,**Oliveria decumbens*CHS, PVA (core), PVP, MD (shell)Acetic acid/n.a.Coaxial225–250 nm(45)FTIR, mechanical properties, in vitro antioxidant and antimicrobialn.d.Antimicrobial wound dressings[144]23CloveCHS, PEOAcetic acid/n.a.Conventional/solution154–189 nm(35–43)FTIR, XRD, LE, swelling, in vitro release, in vitro antibacterial and cytotoxicity, in vivo wound healingn.d.Antimicrobial wound dressings[122]24Eucalyptol (phytochemical)PVA, GLLWater/n.a.Conventional/solution219 nm(30)FTIR, TGA, contact angle, mechanical properties, swelling, LE, antibiofilm activity, in vitro antibacterial, time kill n.d.Antifungal agent delivery[127]25Cinnamaldhehyde (phytochemical)PVA, GLLWater/n.a.Conventional/solution204–279 nm(39–58)FTIR, TGA, DTG, contact angle, LE, in vitro release, in vitro antibacterial and anti-proliferative activityn.d.Antibacterial wound dressings[126]26*Mentha longifolia*PCL, SAn.a./n.a.Conventional/solution/impregnation188 nm(36)FTIR, in vitrocytotoxicity & antibacterialn.d.Antibacterial agent delivery[145]27Lemon balm, dillCGH, CHS (shell)n.a./n.a.Coaxial60–120 nm(20–80) ^#^FTIR, LE, in vitro antimicrobial, in vivo biocompatibilityn.d.Antibacterial wound dressings[146]28*Trachyspermum ammi*PVA, GEL (core), PVP, Ab, AV (shell) Water, ethanol/n.a.Coaxial623 nm(160)FTIR, mechanical properties, porosity, in vitro antioxidant and antimicrobial, in vitro and ex vivo release, in vitro cytotoxicity, in vivo antibacterial and wound healingn.d.Wound healing acceleration[124]29Tea tree, neemPVAEthanol/n.a.Conventional/solution315–355 nm (n.d.)FTIR, DSC, LE, water retention, ex vivo permeation, in vitro antibacterial and cell compatibility, clinical assessmentn.d.Topical treatment of acne[147]30OreganoPVDFDMF/n.a.Conventional/solution620–770 nm(105–141)FTIR, TGA, contact angle, in vitro release, in vitro antioxidant and cytotoxicity6-month antioxidant and MTT (ambient light–dark conditions)Antitumor agent delivery[148]31CinnamonPUDMF, THS/n.a.Conventional/solution179–209 nm(28–34)FTIR, DSC, mechanical properties, swelling, in vitro antibacterial and cytotoxicityn.d.Diabetic ulcer wound healing[149]EO: Essential Oil; SD: standard deviation; n.a.: not applicable; n.d.: no data; ^a^ PAN: polyacrylonitrile, CA: cellulose acetate, SA: sodium alginate, PEO: polyethylene oxide, PVP: polyvinyl pyrrolidone, HP-βCD: hydroxypropyl-β-cyclodextrin, SF: silk fibroin, GEL: gelatin, PVA: polyvinyl alcohol, ZN: zein, PU: polyurethane, PLCL: poly(L-lactide-co-ε-caprolactone), PCL: poly-ε-caprolactone, MD: maltodextrin, GLL: gellan, CGH: collagen hydrolysates, Ab: arabinose, AV: aloe vera, PVDF: polyvinylidene fluoride; ^b^ DMF: dimethylformamide, HFIP: 1,1,1,3,3,3-hexafluoro-2-isopropanol, THF: tetrahydrofuran, GAA: glacial acetic acid, GMA: glycerol monoacetate, DCM: dichloromethane, SDS: sodium dodecyl sulfate; ^c^ TGA: thermogravimetric analysis, FTIR: Fourier-transform infrared, LE: loading efficiency, DSC: differential scanning calorimetry, WVTR: water vapor transmission rate, XRD: x-ray diffraction, MTT: 3-(4,5-dimethylthiazol-2-yl)-2,5-diphenyl-2H-tetrazolium bromide, AFM: atomic force microscopy, APTT: activated partial thromboplastin time, PT: prothrombin time, DTG: derivative thermogravimetry; ^#^ Deduced by bar chart examination.


Heydari–Majd et al. [134] utilized conventional solution ES to encapsulate *Ferula gummosa Boiss* EO into zein (ZN) nanofibers as delivery systems for diabetes control. The topology and size of the EONFs were studied by SEM, possible chemical interactions by FTIR spectroscopy and thermal behavior by DSC. The in vitro antioxidant and the α-amylase and α-glucosidase inhibition activities of the prepared EONFs were also evaluated. SEM images showed that the morphology and fiber size of EONFs were dependent on the ZN concentration used in the fibers. As the ZN concentration increased from 25% to 40%, the mean diameter of EONFs increased from 410 to 788 nm, and the large beads and spindle-like webs changed to bead-free ones. FTIR spectroscopy and DSC studies verified the encapsulation of EO into the nanofibers with loading near 95%. EONFs retained their antioxidant activity in vitro and showed α-glucosidase and α-amylase inhibition with IC50 values ranging from 0.78 to 1.25 and 1.09 to 1.64 mg/mL, respectively. Thus, these EONFs appear promising for the control of diabetes.

Sofi et al. [137] used conventional solution ES to encapsulate lavender EO into PU/silver nanoparticles (AgNPs) composite nanofibers as antibacterial wound dressings. The topology and size of the prepared EONFs were with SEM and TEM, and potential chemical interactions were studied by FTIR spectroscopy and XRD. In addition, the wettability, in vitro antibacterial activity, cytotoxic action and cell attachment ability, were investigated. The authors found that the incorporation of AgNPs decreased the diameter of EONFs, whereas lavender oil increased it. FTIR spectroscopy and XRD confirmed the adequate encapsulation of EO into the nanofibers. The presence of AgNPs and lavender oil resulted in hydrophilicity enhancement of the nanofibers, and thus ensured the in vitro proliferation of chicken embryo fibroblasts. The antibacterial activity of EONFs was studied against *E. coli* and *S. aureus* strains, yielding inhibition zones of 16.2 and 5.9 mm, respectively. The present EONFs dressings were shown to provide protection against external agents and to promote tissue regeneration, thus showing great potential as multifunctional wound dressings.

Liu et al. [143] utilized in situ conventional solution ES to encapsulate thyme EO into ZN/PEO nanofibers as antibacterial wound dressings. The topology and size of the nanofibers were studied by means of SEM, their porosity by gas permeability, their hydrophilicity by contact angle measurements and chemical interactions by FTIR spectroscopy. Furthermore, the in vitro antibacterial activity and in vivo wound healing ability of the EONFs, were investigated. The EONFs showed excellent values of gas permeability (up to 154 m^2^/s) and superhydrophilicity (contact angle ~ 0°). The antibacterial activity of EONFs was studied in vitro, and the results suggest that the in situ EONFs greatly promoted wound healing within 11 days. These results highlight the potential of in situ ES for personalized medicine therapy by direct deposition of EONFs onto the site of the wound to accommodate the individual’s specific wound shape and facilitate its healing with convenience and comfort.

El Fawal and Abu-Serie [148] utilized conventional solution ES to encapsulate oregano EO into PVDF nanofibers as delivery systems with antitumor properties. The topology and size of the nanofibers were studied by means of SEM, their hydrophilicity by contact angle measurements, potential chemical interactions by FTIR spectroscopy and thermal behavior by TGA. The in vitro release, antioxidant activity and cytotoxicity of EONFs were also investigated. Moreover, they carried out a long-term (6-month) stability study at room temperature under light—dark cycle conditions to detect its effects on the antioxidant and cytotoxicity properties of the EONFs. The SEMs taken from the EONFs showed the formation of bead-free homogenous fibers with an average diameter between 620 and 770 nm. FTIR spectroscopy confirmed the encapsulation of the oregano oil into the nanofibers and contact angle measurement confirmed the hydrophobicity of PVDF. Biocompatibility studies on human normal cells (Wi-38) showed safety and apoptosis-mediated anticancer activity of EONFs (showing also strong antioxidant properties). The antioxidant and anticancer activities of the EONFs were still left even after a long-term stability study. Hence, the present EONFs could be promising candidates in anticancer treatment.

### 4.2. Studies Utilizing Emulsion Electrospinning (EES)

Emulsion electrospinning (EES) was developed to overcome the limitations of solution ES related to the use of toxic solvents and low production yields. Table 1 shows that only 16% of the studies utilized EES for the preparation of EONFs. Three of them are discussed in this sub-section as the most indicative ones.

Hajiali et al. [132] utilized EES to encapsulate lavender EO into PEO/SA nanofibers as antimicrobial/anti-inflammatory burn healing dressings. The topology and size of the EONFs were studied by means of SEM, and potential chemical interactions by Raman spectroscopy. The wettability and mechanical properties of EONFs were also investigated. Furthermore, the in vitro release, antibacterial, anti-inflammatory, cytotoxic and in vivo cytokine expression actions were investigated. The EONFs inhibited the growth of *Staphylococcus aureus* and were able to control and reduce the inflammatory responses that were induced in human foreskin fibroblasts by lipopolysaccharides, and in rodents by UVB exposure. The reduction of pro-inflammatory cytokines was also reported. This suggests that the present EONFs are promising and advanced biomedical devices for burn management. Further studies, however, are needed to establish the effectiveness of these nanofibrous dressings on deeper burns.

Rafiq et al. [123] systematically studied the encapsulation of clove, cinnamon or lavender EO into PVA/SA nanofibers by applying Box—Behnken design of experiments (DoE) to optimize the EES process parameters. The EONFs were intended for antibacterial wound dressings. The topology and size of EONFs were studied by means of SEM (the diameter of EONFs was not reported) and potential chemical interactions were studied by FTIR spectroscopy. The water absorption properties and in vitro antibacterial activity of EONFs were also investigated. The EONFs showed high antibacterial activity against *Staphylococcus aureus*. The EONFs loaded with 1.5 % cinnamon oil exhibited the highest inhibition zone of 2.7 cm. The EONF-coated cotton gauze showed higher water absorption capacity compared to simple cotton gauze, and thus had better potential to be used as an antibacterial wound dressing.

More recently, Aman et al. [121] reported a systematical study on the encapsulation of clove EO into PVA nanofibers by applying Taguchi’s DoE for the optimization of EES process parameters. The EONFs were designed as topical delivery scaffolds of anti-inflammatory agents to the skin. The topology and fiber size of EONFs were studied by means of SEM. Potential chemical interactions and thermal behavior were investigated with FTIR spectroscopy and DSC, respectively. Ex vivo skin permeation, in vivo anti-inflammatory studies and skin irritation tests were also performed. In addition, EONFs were subjected to a 6-month stability study according to ICH guidelines (5 °C and ambient temperature) to find out the effects of storage on the retention of EO. FTIR spectroscopy and DSC results confirmed the encapsulation of EO in the EONFs. *Ex vivo* skin permeation data of clove EO showed that it retained its penetration ability through the skin. Topical treatment with EONFs promoted in vivo anti-inflammatory activity against a croton oil-induced mouse skin inflammation model. This was verified by histopathological and immunohistochemical examinations. The present EONFs showed also a high stability upon storage at 5 ± 3 °C for 6 months.

### 4.3. Studies Utilizing Coaxial Electrospinning (CES)

Coaxial ES (CES) is an interesting technology for the preparation of EONFs since we can mainly avoid the limitations and drawbacks associated with conventional solution ES and EES. CES enables the generation of core-sheath nanofibers, where the EO is loaded (and protected) in the core without being mixed with any solvents or surfactants that could be harmful to its chemical consistency. CES has been utilized for the preparation of EONFs intended for pharmaceutical or biomedical applications only during a recent two-year period (2021–2022). Thus, it is still an understudied method, although very promising technology for the preparation of EONFs. As seen in Table 1, only three studies were traced in the literature utilizing CES for the preparation of EONFs. These studies are discussed in more detail in this sub-section.

Barzegar et al. [144] used CES to encapsulate *Satureja mutica* or *Oliveria decumbens* EO into PVP/maltodextrin (MD) core-shell nanofibers as antimicrobial wound dressings. The topology and size of EONFs were studied by means of SEM and potential chemical interactions were investigated by FTIR spectroscopy. The mechanical properties of EONFs were also evaluated. Moreover, in vitro antioxidant and antimicrobial activities of EONFs were studied. SEM images showed that EONFs were bead-free with an average diameter between 225 and 250 nm. EONFs showed good antioxidant and antimicrobial activities (against *P. aeruginosa*, *E. coli*, *S. aureus*, *C. dubliniensis* and *C. albicans*). The results suggest that the core-sheath EONFs are obviously applicable as dressings for dry wounds. For moist wounds, a suitable crosslinking treatment for the stabilization of EONFs is needed, and therefore, further studies are proposed to be performed.

Râpă et al. [146] used CES to encapsulate lemon balm and/or dill EO into CHS/collagen hydrolysates (CGH) shelled nanofibers as antimicrobial wound dressings. The topology and size of EONFs were studied by means of SEM and potential chemical interactions were investigated by FTIR spectroscopy. The in vitro antimicrobial activity and in vivo biocompatibility were also examined. In vitro results showed a synergistic effect of EONFs loaded with dill and lemon balm EOs on the antimicrobial activity (against *S. aureus*, *E. faecalis*, *C. albicans* and *C. glabrata*) compared to EONFs loaded with either dill or lemon balm oil alone. The in vivo test results suggested good biocompatibility of the EONFs, mainly due to the presence of CGH in the shell of the nanofibers, thus making them suitable for wound dressings.

Zare et al. [124] studied semi-systematically the encapsulation of *Trachyspermum ammi* EO into PVA/GEL shelled nanofibers by screening different combinations of core (PVP, arabinose and aloe vera) and shell (PVA and GEL) polymer concentrations to optimize the CES process. The core-sheath EONFs were designed as wound dressings that could accelerate a wound healing process. The topology and size of EONFs were studied by means of SEM and potential chemical interactions were investigated by FTIR spectroscopy. The porosity and mechanical properties of EONFs were also studied. Moreover, the in vitro antioxidant and antimicrobial activities, in vitro/ex vivo release properties, and in vitro antibacterial and wound healing properties, were also investigated. SEM images demonstrated that the core-sheath structured EONFs had an average diameter of between 564 and 746 nm. EONGs showed excellent in vitro antimicrobial and antioxidant activity. In vitro and ex vivo release studies revealed a prolonged release profile of the EO from the EONFs. Furthermore, in vivo antibacterial activity, wound healing and histomorphological examinations confirmed the high efficacy of the EONFs in the treatment of *S. aureus* infected full-thickness rat wounds. Thus, the present core-sheath EONFs show good potential as novel wound dressings for skin injuries.

### 4.4. Challenges and Outlooks in the Fabrication of Essential Oil-Loaded Nanofibers

It is well known that the physicochemical, mechanical and pharmaceutical properties of polymeric nanofibers are size-dependent. Since nanoscale fiber dimensions can directly impact surface-free energy, the instability of such nanofibers under long-term storage can be often recognized in the literature. Moreover, EOs are by their nature physically and chemically unstable, as already discussed. Thus, an EONF system is expected to present some instability under ambient or extreme (high temperature/relative humidity, refrigeration) conditions. The EONFs intended for biomedical applications are fabricated with biomimetic polymers to enhance cell adhesion, proliferation and viability. However, these systems exhibit poor mechanical properties, since EOs have elastic properties, thus confining the mechanical properties of nanofibers. Moreover, EONFs exhibit a fast degradation rate, and complete scaffold disintegration when immersed in simulated biological fluids. Thus, the fabrication of EONFs with robust characteristics and long-term stability is mandatory for biomedical and pharmaceutical applications. Furthermore, EOs can have long-term effects on the physicochemical properties of the EONFs. EOs can act as plasticizers for some polymers (e.g., PVA), thus resulting in a reduction in the glass-transition temperature (Tg) of the corresponding EONFs. To date, little is known about the effects of EOs on the aging of EONFs, and presumably, if EOs could induce changes in the morphology, thermal and mechanical stability and degradation rate of the EONFs. Surprisingly, only four out of the total 31 studies (included in this review) investigated the long-term stability of EONFs. Therefore, there is a great gap in the literature, which hampers the commercialization of such systems.

One additional aspect to be considered in future research works is the impact of potential changes in the chemical consistency of EOs during ES on the final properties of EONFs. Such process-induced consistency changes in EOs could be due to solvent evaporation or utilization of surfactants. To overcome these drawbacks, CES can be evidently used instead of conventional solution ES or EES. In CES, the EO can be incorporated into the core-sheath nanofiber by electrospinning it either alone or dispersed in non-interacting polymers, thus protecting it from contact with unsuitable solvents or surfactants. CES for the fabrication of EONFs is still new in the pharmaceutical and biomedical field, and more research work is urgently needed to fully explore its potential for the fabrication of robust and long-term stable EONFs.

## 5. Summary and Conclusions

Electrospinning (ES) is a promising technology for the successful encapsulation of essential oils (EOs) into biocompatible and biomimetic polymeric nanofibers intended for pharmaceutical and biomedical applications. There is a number of studies in the literature reporting the fabrication of EO-loaded nanofibers (EONFs) by means of a conventional solution ES method. However, the application of a solution ES method limits the need for toxic solvents in the process, and consequently, arising environmental concerns. Moreover, this process may alter the chemical consistency of EOs, and it is characterized by a low production yield. Emulsion ES (EES) is an alternative technology to conventional ES, but this method may negatively affect EOs due to the interfacial tension between the two immiscible phases of the emulsion. Coaxial ES (CES) is a more recently introduced technology for the preparation of EONFs for pharmaceutical and biomedical applications. CES appears to be a very promising method, and it is free of the limitations/drawbacks associated with conventional solution ES and EES. Hence, future research work is needed to unravel the potential of CES. Moreover, the present review highlights the absence of long-term stability data of the EONFs in the literature, thus hindering the commercialization of such drug delivery systems or biomedical constructs. EONFs hold promises especially for the systematic or topical delivery of natural antimicrobial agents and for the preparation of biomaterials for biomedical applications. The lack of systematic studies on their performance in vivo, however, will delay their appearance on the market.

## Figures and Tables

**Figure 1 pharmaceutics-14-01799-f001:**
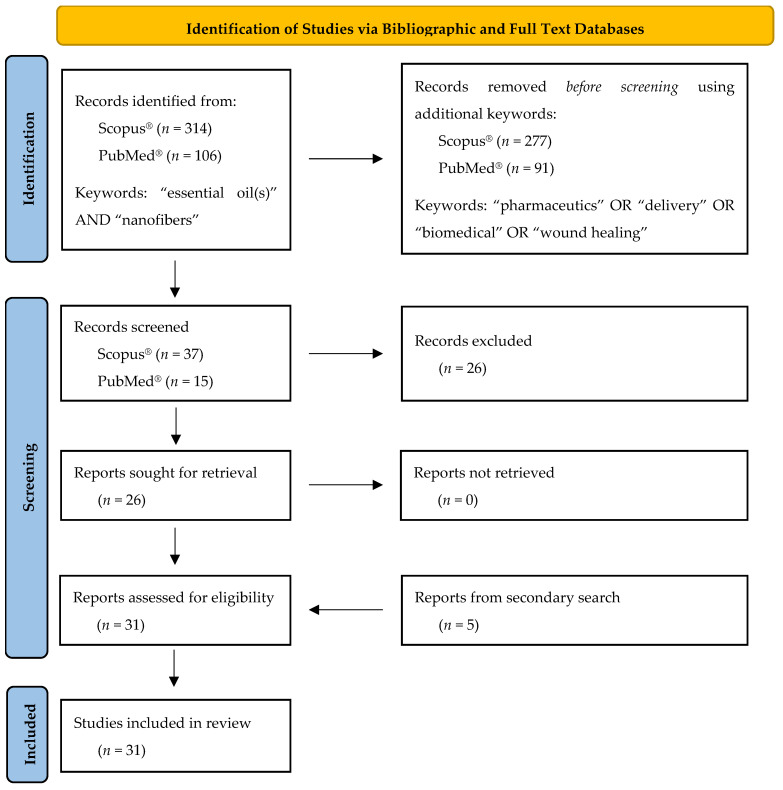
Flow diagram describing the selection, identification and screening methodology of the studies of interest included in the present review according to PRISMA guidelines [25].

**Figure 2 pharmaceutics-14-01799-f002:**
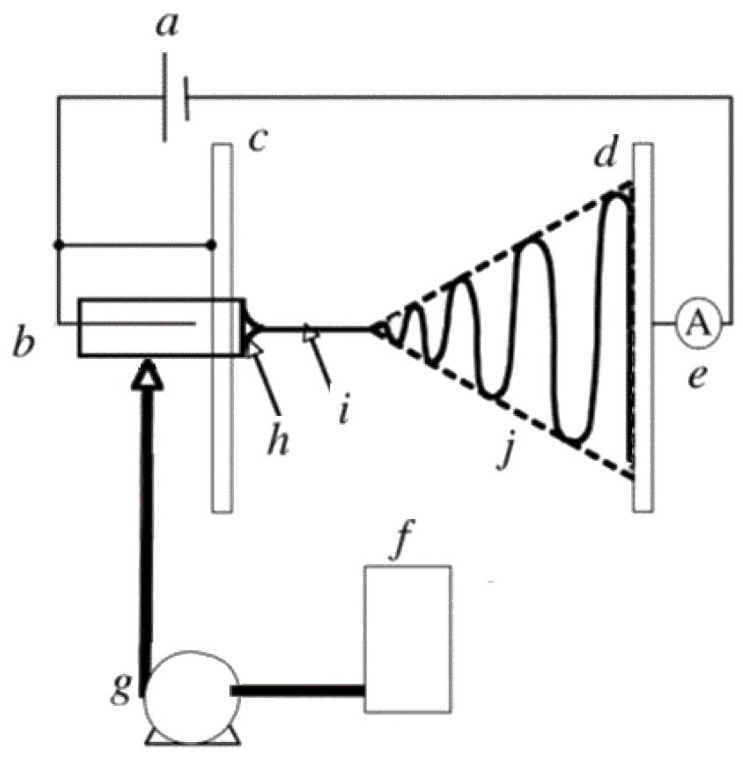
Schematic representation of a conventional ES set-up: (**a**) high-voltage supply, (**b**) charging device of the spinneret, (**c**) high-potential electrode, (**d**) grounded collector, (**e**) current measurement device, (**f**) working solution reservoir, (**g**) flow rate control of the syringe pump, (**h**) Taylor’s cone, (**i**) thinning jet and (**j**) instability region. Reprinted with permission from [29].

**Figure 3 pharmaceutics-14-01799-f003:**
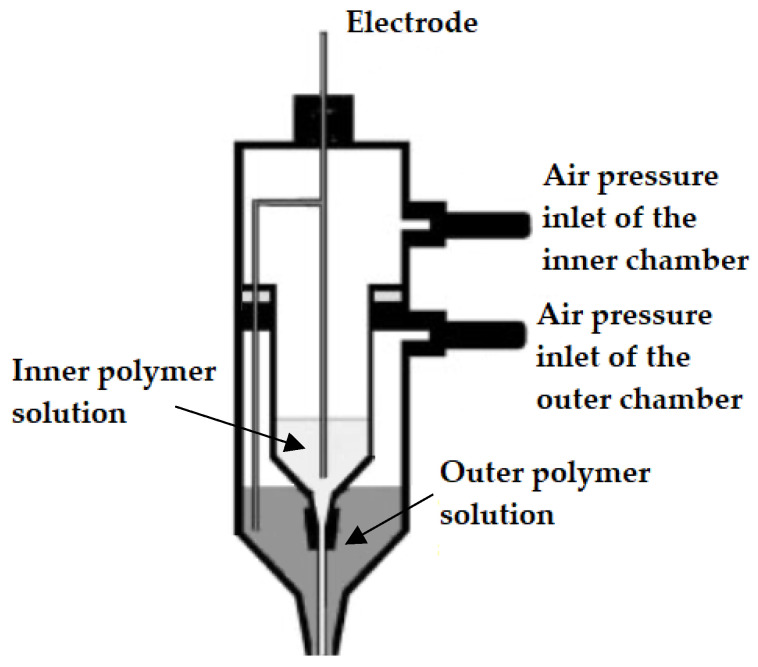
Original experimental set-up used for the first time in a coaxial electrospinning process (CES). Adapted with permission from [37].

**Figure 4 pharmaceutics-14-01799-f004:**
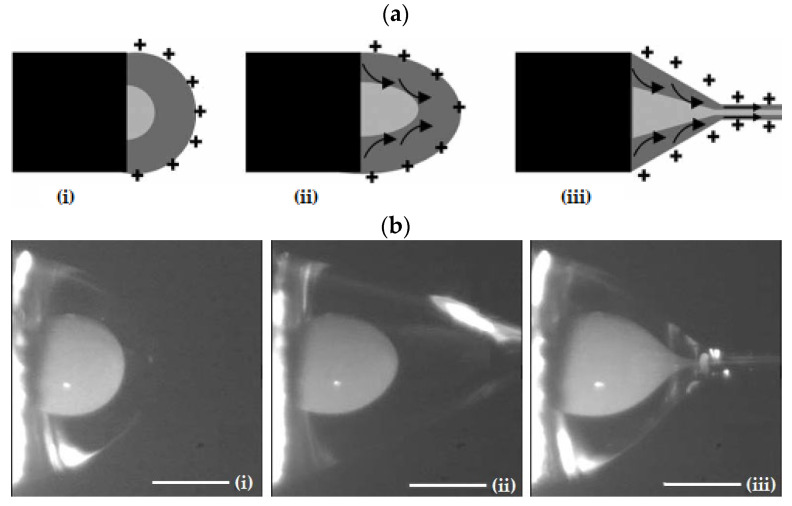
(**a**) Schematic illustration of the formation of a compound Taylor cone: (**i**) Surface charges on the sheath solution, (**ii**) Viscous drag exerted on the core by the deformed sheath droplet, (**iii**) Sheath-core compound Taylor cone formed due to continuous viscous drag). Adapted with permission from [41]. (**b**) Formation of compound Taylor cone menisci and the electrified coaxial jet. Voltage increases from (**i**) to (**iii**) (scale bar 0.5 mm). Adapted with permission (CC BY 4.0) from [42].

**Figure 5 pharmaceutics-14-01799-f005:**
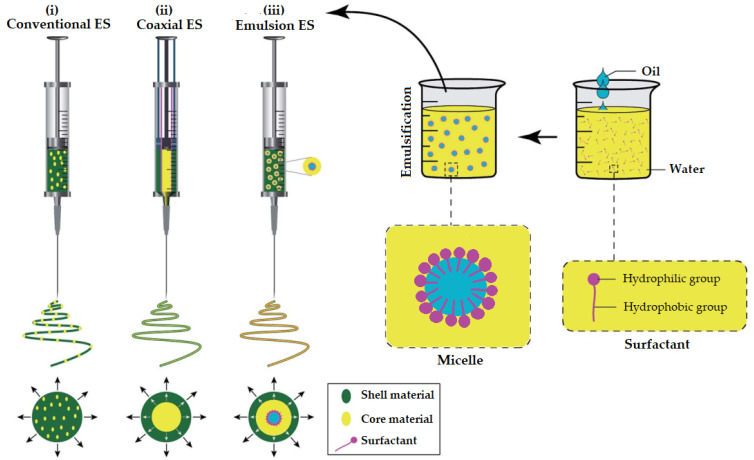
Schematic illustration of spinnerets loaded with different spinning liquids for (**i**) conventional (traditional), (**ii**) coaxial and (**iii**) emulsion electrospinning (ES). Adapted with permission (CC BY 4.0) from [47].

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
