# Peer review of "Essential Oil—Loaded Nanofibers for Pharmaceutical and Biomedical Applications: A Systematic Mini-Review"

_pharmaceutics, 2022, doi:10.3390/pharmaceutics14091799_

Round 1
Reviewer 1 Report
This review is an interesting manuscript for essential oil–loaded nanofibers. I believe that touches all the important topics, and is well written, with scientific soundness and methodology described, I think the manuscript fulfills all the necessary elements to be published in this important journal. just a couple of comments are given
a) Should be a good idea to separate or differentiate between natural and synthetic polymers in section 3.3, because most of the natural polymers you presented are difficult to electrospun by themselves
b) In section 3.3., are just the enlisted polymers the uniques for the preparation of EONFs?, if can be others please state
Reviewer 2 Report
The present review overviews the potential use of Essential Oil loaded Nanofibers in the pharmaceutical and biomedical fields. I congratulate the authors for the presented work which I consider to be well developed in terms of research method, and also on the topics organisation. The manuscript is logical and very readable, and reviews a series of adequate references related to the theme, with information summarised in a proper series of figures and tables. Thus, I recommend the manuscript for publication.
Reviewer 3 Report
Dear authors,
The paper entitled Essential Oil – Loaded Nanofibers for Pharmaceutical and Biomedical Applications: a Systematic Mini-Review, is very interesting and well documented and organized.
Only minor corrections should be made:
Line 181 – removed Schematic illustration, which is write two times
Line 409 , 416– use the same name: Candida albicans versus C. albicans
At the beginning of 4.1 Studies Utilizing Conventional Solution Electrospinning (ES), the authors must present the criteria which were use to select the presented detailed studies: Tonglairoum et al., Heydari–Majd et al. Sofi et al., Liu et al., El Fawal and Abu-Serie, considering that in Table 1 more references are listed.
4 The same recommendation for 4.2 Studies Utilizing Emulsion Electrospinning (EES)
Reviewer 4 Report
Please see attached.
